# Yeṣer ha-Raʿ and Original Sin

**Matthew Wade Umbarger** 

School of Catholic Studies, Newman University, Wichita, KS 67213, USA; umbargerm@newmanu.edu

**Abstract:** Many modern rabbis insist that original sin was invented by St. Paul, and that it does not have a Jewish antecedent. Instead, rabbinic Judaism explains human evil in terms of "*yeṣer ha-raʿ*," "the evil inclination." But evidence from Second Temple period wisdom and apocalyptic literature suggests that ideas like Paul's were indeed common in certain quarters of Jewish thought in the first century. Paul's doctrine of original sin draws from an assortment of Old Testament texts. What seems novel in Romans 5 is essentially an aspect of his apocalyptic vision. Rabbinic texts from the Mishnah onwards intentionally suppress this apocalyptic account of original sin. Instead of original sin, rabbinic doctrine posits the *yeṣer ha-raʿ* as the explanation for human wickedness. This is an innate aspect of human nature. But it is something that good discipline, and especially the practice of Torah, can amend. Some aspects of Pauline teaching actually run parallel to these later texts pertaining to the *yeṣer ha-raʿ*, as well. In particular, his use of *sarx* seems to be a theological cognate to this concept of an evil inclination.

**Keywords:** *yeṣer ha-raʿ*; original sin; apocalypticism; Paul; *sarx*

## 1. Introduction

A number of years ago, my friend Rabbi Nissim Wernick (זכרונו לברכה) invited me to have lunch with him, and in between his breadsticks and soup, proceeded to pelt me with Jewish criticisms of Christianity. The major theme of his diatribe was that Christianity, specifically the Apostle Paul, had invented the doctrine of original sin so as to justify construing the Messiah into a spiritual redeemer for all of humankind, rather than a political redeemer for Israel. Rabbinic Judaism, going all the way back to the days of the Pharisees, he assured me, was completely unaware of such a doctrine. It was an intriguing argument, and one for which I was ill-prepared to counter (much to Rabbi Wernick's delight, I assure you). Since then, I have confirmed that this is indeed the mainstream view of rabbinic Judaism. For instance, in his influential book from the 1950s, *Where Judaism Differed*, Abba Hillel Silver wrote that "Jewish theology accepts no . . . doctrine of man's corrupt origin, 'that all men descended from Adam contract original sin from him, and that this sin is transmitted by way of origin.'" ([Silver 1957](), pp. 158–59).

I have been haunted by Rabbi Wernick's challenge ever since that last meal with him. His criticisms niggle at me primarily for two reasons. First of all, one of my great joys in researching early Christian belief and practice has been to discover that almost all of it developed organically from Jewish doctrine and ritual. (Rabbi Wernick was always quick to acknowledge this.) If what Rabbi Wernick was saying was true, then here was a glaring instance in which a fundamental Christian doctrine did *not* develop from a Jewish precedent. Obviously, the Holy Spirit could have revealed this to the early Church, or to Paul as a part of the new dispensation of Grace, but Paul does not speak of original sin like this. Typically, when he takes on teaching something that he has received as a new revelation, he uses words like "mystery". This is not the case in the fifth chapter of Romans. So, why does Paul seem to assume that his audience will already know what he is talking about if he is making this doctrine up on the spot?

I was also recalling the years that I had studied at Ozark Christian College, where my professors had instructed me in the majority position of the Cambpellite Restoration

Movement: original sin was not really a biblical doctrine.[1] Rather, each human individual has learned to sin by the example of the world at large. (For this reason, infant baptism was illegitimate.) One professor had gone so far as to depict Augustine as a heretic opposed by his "orthodox" champion, Pelagius. To some extent, Rabbi Wernick seemed to be in agreement with them. When I made the decision to become Catholic, it meant that I had to abandon this specific doctrinal position. It bothered me that my former co-religionists might share something in common with Judaism that I could not.

So, in this short article, I want to revisit this discussion with Judaism about original sin by suggesting that Paul did indeed develop his ideas, guided as he was by the Holy Spirit, from older doctrines that can be ascertained from Second Temple period literature, especially apocalyptic texts. We will see that Paul's doctrine of original sin is essentially an aspect of his apocalyptic vision. I will also propose that Paul's doctrine of the *sarx*, the flesh, runs parallel with certain teachings that were retained in rabbinic Judaism pertaining to the *yetzer ha-ra*, the evil inclination, which is often posited as a sort of Jewish alternative to original sin.

But first, a few caveats. I am intentionally avoiding inter-denominational squabbles about original sin within the Church. Thus, as important as the disputes between East and West about original and/or ancestral sin are, this article will perhaps sound as though I believe that they have all been resolved, or never amounted to much to begin with. This is certainly not the case. Nor am I delivering this as a rebuttal of all of the modern, Christian rejections of the doctrine of original sin, although it can probably be construed that way. And, for the sake of simplicity, I am going to refer to Paul's doctrine of the consequences of the Fall as "original sin", although many Christians today would surely balk at that choice of vocabulary.[2]

## 2. Original Sin in Romans Five

Let's begin with the source-text in Paul, in Romans five, as rendered in the RSV.

12 Therefore as sin came into the world through one man and death through sin, and so death spread to all men because all men sinned—13 sin indeed was in the world before the law was given, but sin is not counted where there is no law[3] . . . 18 Then as one man's trespass[4] led to condemnation for all men, so one man's act of righteousness leads to acquittal and life for all men. 19 For as by one man's disobedience many were made sinners, so by one man's obedience many will be made righteous.

As Fitzmeyer remarks, "Adam's disobedience placed the mass of humanity in a condition of sin and estrangement from God; the text does not imply that they became sinners merely by imitating Adam's transgression; rather, they were constituted sinners by him and his act of disobedience."[5]

There are, of course, numerous problems of interpretation in these verses that Christians have been arguing over for centuries, including Verse 12, "the *crux* of this difficult passage", (Sanday and Headlam 1895, p. 134) particularly over the phrase translated "because all men sinned" by the RSV, "ἐφ' ᾧ πάντες ἥμαρτον". Origen seems to have misinterpreted the phrase in construing it as meaning that everyone has sinned *in* Adam, and Augustine caused this to be the standard view in Western Christianity (however, it is possible to "distinguish between acceptance of Augustine's general understanding of the thought of the clause and acceptance of his grammatical explanation of 'in quo'", as Cranfield noted) (Cranfield 1975, p. 276). The translation of the RSV is probably correct, and yet, as Sanday and Headlam pointed out, this does not necessarily sever the "the connexion between Adam and his posterity. If they sinned, their sin was due in part to tendencies inherited from Adam." (Sanday and Headlam 1895, p. 134).



### 3. Original Sin in Rabbinic Judaism

Nearly all modern rabbis seem to agree that here in Paul there are fundamental differences with their own theological anthropology, insisting that there is nothing in the Jewish literary corpus that has any hint of correlation with Paul's statements here. To put it succinctly, for rabbinic Judaism, Adam's trespass led to his own personal condemnation, full stop, and not the condemnation of all humankind.[6]

There are exceptions, of course. For instance, Rabbi David Kimchi, in his commentary on Isaiah 43:27, "Your first father sinned", wrote, "And how will you say that you have not sinned, when, behold, your first father sinned, the first man, because Adam was stamped with sin, because 'the inclination of a man's [or Adam's] heart is evil from his youth (Genesis 6:5)?'"[7] Samuel S. Cohon, representing the Reform Jewish tradition, said that the doctrine of original sin "in varying forms figures in Jewish as in Christian thought" and that it "derives its vitality from the raw facts of life." (Cohon 1987, p. 219). He argued, as I will here, that when Christianity developed this dogma, it was "following certain trends in Judaism", though he acknowledged that Judaism never assigned "to it the importance which it occupies in Christianity". Cohon also denied that Genesis three has anything to do with original sin, "contrary go the uses made of it by Paul and his followers", though he acknowledged that it serves as an etiological explanation of our mortality (Cohon 1987, p. 220). He explained, "The Yahvist concerns himself with the origin of death and suffering rather than with the origin of human sinfulness". Thus, sin is "a power external to man." (Cohon 1987, p. 225). Most importantly, he insisted that any notions of sin imputed to Adam's progeny because of his fall put forth "by both Christianity and Judaism are without foundation."[8]

### 4. Source Texts from the Hebrew Bible

Surely, Paul's ideas about original sin, including his exegesis of Genesis three, are not completely original, albeit they may be more clear and refined than many of the other texts that I am about to bring up for our consideration. Most of these are from the Second Temple period or a bit later, but even in the Hebrew Bible there are abundant sources for Paul's thought. In polemical proof-texting, apologists often appeal to Psalm 51:5, of course: "Behold, I was brought forth in iniquity, and in sin did my mother conceive me."[9] But I think that Paul was influenced more by a Torah tradition stretching from the words of Exodus 34:7, (where God is said to visit "the iniquity of the fathers upon the children and the children's children, to the third and the fourth generation")[10], through the prayers of identificational repentance in Nehemiah 1 ("I and my father's house have sinned," v. 6) and Daniel 9 ("for our sins, and for the iniquities of our fathers, Jerusalem and thy people have become a byword among all who are round about us," v. 16). An even more proximate theme that Paul is borrowing from the Hebrew Bible is that of the concrete, microcosmic representative of the nation deciding the fate of his people, especially prominent in the Deuteronomistic history of the kings of Israel. For instance, 2 Kings 24:3 drily observes that Jerusalem suffered all of her disasters at the hands of Babylon "for the sins of Manas'seh", beginning in the reign of Jehoiakim, Manasseh's grandson. This very idea was taken up by Jeremiah, in his message that judgement for Judah was inevitable: the Lord will make the people of Judah a horror "because of what Manas'seh the son of Hezeki'ah, king of Judah, did in Jerusalem" (15:4).

The prophetic vision of divine justice is simply not democratic. We can indeed suffer terrible judgements when those who represent us before God sin against Him. For Paul, Adam was the Deuteronomistic king *par excellence*, representing all of humanity, and his sin resulted in our bitter exile of estrangement from God, and consequently, only the Second Adam could restore us to the place of divine honor.

### 5. Original Sin in Other Second Temple Period Texts

Paul was not the only Second Temple period Jewish author to synthesize all of this biblical material in such a way. Samuel S. Cohon recognized this, writing, "Only in



Apocryphal and Pseudepigraphic Jewish writings does the Paradise story begin to figure as the basis for speculation regarding the origin of death and of sin." (Cohon 1987, p. 228). And C. E. B. Cranfield went even further, arguing that "what is implicit in the OT account [of Genesis three] was of course, made fully explicit in later Jewish writings." (Cranfield 1975, p. 280). He also suggested that it is probable that "Paul was familiar with many of the ideas concerning Adam to be found in the Apocrypha and Pseudepigrapha and in Rabbinic literature", although "the restraint and sobriety of his own references to Adam are noticeable." (Cranfield 1975, p. 281). His explanation for this restraint is Paul's motive to focus on the person and work of Jesus rather than Adam. "Adam in his universal effectiveness for ruin is the type which—in God's design—prefigures Christ in His universal effectiveness for salvation." (Cranfield 1975, p. 283). James Dunn concurs: "Paul here shows himself familiar with and indeed to be a participant in what was evidently a very vigorous strand of contemporary Jewish thinking about Adam and the origin of evil and death in the world."[11]

Ben-Sira has this to say about Eve, in the first half of the second century BC: "From a woman sin had its beginning, and because of her we all die" (Ecclesiasticus 25:24). True, Ben-Sira throws this out as but one among a whole gob of rather chauvinist proverbs that show up in this chapter. (One can't help but suspect that he did not have a very happy home life.) Cohon pointed out that in Ben-Sira, "this idea is completely isolated, and contrasts with the general trend of the book to regard morality as a law from everlasting." (Cohon 1987, p. 228). But all the same, the proverb works on the assumption that the sin of Adam and Eve engendered our own transgressions and deaths.

Approximately one hundred years prior to Paul writing his Epistle to the Romans, the author of the second chapter of the Wisdom of Solomon provided a similar, less chauvinistic etiology for our mortality, and introduces the devil into the picture: "23 God created man for incorruption, and made him in the image of his own eternity, 24 but through the devil's envy death entered the world, and those who belong to his party experience it". Paul actually echoes this text in the wording of Romans 5:12 (Cranfield 1975, p. 274).

Going in the other direction, about 100 years after Paul, virtually the same sentiment is contained in the twenty-third chapter of the Apocalypse of Abraham. Similarly, in the Greek Apocalypse of Moses 32, Eve, lamenting her transgression, cries out, "all sin has come about in creation through me!"[12] So, these ancient Jewish sources are aware of the interpretation that the serpent was not just a snake, but at the very least an agent of Satan, bringing about the current human condition of sin and death. And this dualistic version of the Fall, where the choice of Adam and Eve constitutes a cosmic battlefield for good or for evil, is where apocalypticism enters in.

**6. Original Sin in 2 Baruch**

From about the same time, 2 Baruch takes it for granted that we suffer death as a result of Adam's sin, indeed, as a penalty. Consider this section from chapter fifty-six, where the author describes a vision of dark waters that is as terrifying a description of original sin as you could hope for:

> And as you first saw the black waters on the top of the cloud which first came down upon the earth; this is the transgression which Adam, the first man, committed. For when he transgressed, untimely death came into being, mourning was mentioned, affliction was prepared, illness was created, labor accomplished, pride began to come into existence, the realm of death began to ask to be renewed with blood, the conception of children came about, the passion of the parents was produced, the loftiness of man was humiliated, and goodness vanished (2 Baruch 56: 5–6).[13]

For a second century Jewish text, there is a whole lot here that anticipates the later Christian descriptions of the consequences of Adam's fall, including the hint that somehow it is transmitted through the "passion of the parents" as later asserted by the likes of Augus-

tine and Maximus the Confessor. Klijn observes that though "it is, of course, impossible to prove dependency on" any New Testament texts in 2 Baruch, "the parallels are especially striking with the Pauline Epistles, in particular Romans and 1 and 2 Corinthians."[14]

But 2 Baruch does not place the burden of sin squarely on Adam's shoulders. Chapter fifty-four tempers this perspective, especially. "For even though Adam sinned first, and premature death came upon all, even so, for these that were born from him, each individual has brought future torment to their own soul, or each of them has brought for themselves future acclaim" (verse 15). Then, just a few verses later, the author makes this case even more strongly: "Adam did not bring all of this about except for his own soul, alone. But we have all, each individual, become an Adam unto our own soul" (verse 19).[15]

### 7. Original Sin in 4 Ezra

4 Ezra, which was probably written towards the end of the first century, is perhaps the most perspicuous in its presentation of the doctrine of original sin. In chapter seven, Ezra is concluding an exceedingly lengthy debate with God about His justice towards humankind.

> This is my first and last word: It would have been better if the earth had not produced Adam, or else, when it had produced him, had restrained him from sinning. For what good is it to all that they live in sorrow now and expect punishment after death? O Adam, what have you done? For though it was you who sinned, the fall was not yours alone, but ours also who are your descendants. (vss. 116–126).[16]

This is not from a Christian vision of sin and redemption. The verses that follow prescribe the Law of Moses, not the Cross of Christ, as the antidote for these moral ills! So, this is a thoroughly Jewish text.[17]

All the same, 4 Ezra provides a fairly cynical perspective on anyone's capacity to fulfill the requirements of Torah. Ultimately, anyone who is saved must throw themselves upon the grace of God.

> But though our fathers received the law, they did not keep it, and did not observe the statutes; yet the fruit of the law did not perish -- for it could not, because it was thine. Yet those who received it perished, because they did not keep what had been sown in them. And behold, it is the rule that, when the ground has received seed, or the sea a ship, or any dish food or drink, and when it happens that what was sown or what was launched or what was put in is destroyed, they are destroyed, but the things that held them remain; yet with us it has not been so. For we who have received the law and sinned will perish, as well as our heart which received it; the law, however, does not perish but remains in its glory (9:32–37).[18]

Samuel S. Cohon observes, "In his admission of the insufficiency of the Law as the means of redemption, IV Ezra dangerously approaches the Paulinian position." (Cohon 1987, p. 233).

There are other places in 4 Ezra that confirm that the author does indeed intend to put forth some sort of doctrine of original sin. In the third chapter, for instance, he writes that "the first Adam, burdened with an evil heart, transgressed and was overcome, as were also all who were descended from him. Thus the disease became permanent; the law was in the people's heart along with the evil root, but what was good departed, and the evil remained" (vss. 21–22).[19] Consequently, Adam's evil heart now belongs to his descendants, as well (vs. 26, as well as 7:48). These lines also seem to anticipate the rabbinic doctrine of the evil inclination.

Metzger summarizes 4 Ezra's perspective on original sin like this: "This defection is due, in some way, to the sin of Adam (7:[118]), who possessed an evil heart (*cor malignum*, 3:20) in which a grain of evil seed (*granum seminis mali*) had been sown (4:30). Since all of Adam's descendants have followed his example in clothing themselves with an evil heart

(3:26), each is morally responsible. It will be seen that this view corresponds to the rabbinic doctrine of the evil inclination or impulse (*yeṣer ha-raʿ*)."[20]

## 8. Original Sin in Later Midrash

Vestiges of a more crude accounting for original sin exist in certain midrashic texts. For instance, in Yevamot 103b Eve's partaking of the fruit of the tree of knowledge is considered to be a euphemism. In reality, she engaged in coitus with the serpent, and this corrupted her progeny in a morally genetic way. "As Rabbi Yochanan said: 'In the hour when the serpent came upon Eve he cast filth into her. When Israel stood upon Mount Sinai their filth was cut off. Gentiles, who never stood upon Mount Sinai, did not have their filth cut off."[21] It is easy to be distracted by the odd details in this text, but perhaps the most important message here is that Torah is completely efficacious in eradicating original sin.[22] In fact, Alan Cooper writes that "the polemic intent of that text has long been recognized." (Cooper 2004, p. 446).

Similar traditions pertaining to Eve's role in the Fall are recorded in Bereishit Rabbah 17:8. This is a sort of catechism on peculiar differences between men and women, but it concludes with a string of indictments against Eve.

> "And why does the man go out with his head uncovered and the woman with her head covered?"
>
> [Rabbi Joshua] said to them, "Because a transgressor is ashamed before the sons of Adam, so she goes out with her head covered".
>
> "And why do women go walking in front of the dead?"
>
> He said to them, "Because they brought death to the world, so they go walking in front of the dead, as it is written, 'all men follow after him,' (Job 21:33)".
>
> "And why is the commandment of menstrual impurity given to her?"
>
> "Because she poured out the blood of the first man, therefore the commandment of menstrual impurity is given to her".
>
> "And why is the commandment of the challah given to her?"
>
> "Because she brought a curse upon the first man who was the final 'challah offering' of the world, that's why the commandment of the challah was given to her".
>
> "And why was the commandment of the Sabbath candle given to her?"
>
> He said to them, "Because she extinguished the soul of the first man, that's why the commandment of the Sabbath candle was given to her."[23]

Curiously, although for the most part rabbinic literature avoids the subject of original sin, in the sixteenth century a gnostic version of the dogma emerged in the teaching of Isaac Luria. In Lurianic Kabbalah, Adam's transgression has the effect of trapping human spirits in the material world in which they cannot help but sin.[24]

To be sure, there are other etiologies for evil in ancient Jewish literature. So, for example, the Enoch traditions propose that the sons of God who seduced the daughters of men in Genesis six are to blame for almost all of our wicked ways. And the Dead Sea Scrolls, particularly in the *Community Rule*, ascribe evil to God's sovereign decision, in which he assigns some individuals to be ruled over by a spirit of deceit.

## 9. Original Sin an Apocalyptic Doctrine

In some apocalyptic traditions, at least, an account of original sin as having sprouted up in the Garden of Eden is quite evident, and this seems to be what Paul is drawing from in Romans. As is well known, Jewish apocalypticism owes a great deal to exposure to Persian thought. Samuel S. Cohon includes the doctrine of human depravity in the collection of dualistic ideas that exerted influence on early Judaism, and notes that it posed a particular difficulty for Judaism's monotheistic framework.[25]

But Paul does not owe only his doctrine of the fall to this apocalyptic framework. His teachings on the destiny of humanity redeemed from this fall are also apocalyptic. In Romans 5:17, Paul writes, "If, because of one man's trespass, death reigned through that one man, much more will those who receive the abundance of grace and the free gift of righteousness reign in life through the one man Jesus Christ".

This echoes an apocalyptic expectation expressed in Daniel 7:22 and 27 that in the eschaton God's holy ones will receive an everlasting kingdom (See Dunn 1988, p. 282). It is also found in the Psalms of Solomon 3:12: "Those who fear the Lord will rise up into life eternal and their life will be in the light of the Lord and it will never again be eclipsed."[26]

So, both Paul's doctrine of original sin and the eschatological reign of the holy ones who have been redeemed from it are part of a larger corpus of Jewish *apocalyptic* literature. What is unique to Paul is his proposal that Jesus provides the redemption necessitated by the Fall. Consequently, I am convinced that we must approach Romans five as primarily an *apocalyptic* text. Most commentaries on Romans five do the opposite, treating it as a piece of systematic theology that is drawing upon a few ideas that it shares in common with a handful of apocalyptic texts.

### 10. The Yeṣer Ha-Ra'

Rabbinic sources, following the codification of the Mishnah around 200 AD, seem to be virtually oblivious to all of this apocalyptic material. Instead, the source of evil is explained by the *yeṣer ha-ra'*, usually translated "the evil inclination". There actually seems to be an early instance of this doctrine in Ben-Sira. It is especially clear in 15:14: "It was he who created man in the beginning and he left him in the power of his own inclination". A similar idea occurs in 17:31: "What is brighter than the sun? Yet its light fails. So flesh and blood devise evil". Finally, it's possible that 21:11 makes reference to the *yeṣer*: "Whoever keeps the law controls his thoughts, and wisdom is the fulfilment of the fear of the Lord."[27]

However, most rabbinic discussions of the *yeṣer ha-ra'* are based on this mishnah, Berakhot 9:5: "And you shall love the LORD your God with all of your heart . . . . i.e., with both of your inclinations, with the good inclination (*yeṣer ṭōb*) and with the evil inclination (*yeṣer ha-ra'*)". This mishnah is probably derived from the fact that in 1 Chronicles 28:9, the plural for "heart" appears in conjunction with *yeṣer*: "the Lord searches all **hearts**, and understands every **plan** (*yeṣer*) and thought."[28] No doubt, the idea of loving God with our evil inclination strikes Christians as odd. Another text from the Mishnah, Pirqei Avot 4:1 might help: "Who is the mighty one? Whoever subdues their inclination. As it is said, (Proverbs 16:32) 'He who is slow to anger is better than the mighty, and he who rules his spirit than he who takes a city.'"

At times the *yeṣer ha-ra'* is identified with Satan himself. One such example is in Bava Batra 16a:

> Reish Lakish said, "Satan is the *yeṣer ha-ra'*, he is the angel of death, and he is that Satan of whom it is written, 'and Satan went out from before the Lord' (Job 2:7). He is the *yeṣer ha-ra'*, as it is written there, '**only** evil all the day' (Genesis 6:5), and it is written here, '**only** do not stretch forth your hand against him' (Job 1:12). He is the angel of death, as it is written, '**only** spare his soul' (Job 2:6). Apparently Job is in his hands.".
>
> (See Cohon 1987, p. 247)

For Reish Lakish, the recurrence of "only" suggests a deeper connection between these texts, explained by the activity of Satan lurking in the shadows of these verses.

### 11. The *Yeṣer Ha-Ra'* as a "Necessary Evil"

But other rabbinic texts treat the *yeṣer ha-ra'* more ambiguously, as a necessary evil, or even a good. The Zohar goes so far as to say that "the *yeṣer ha-ra'* is as necessary for the world as rain is for the world, for without the *yeṣer ha-ra'*, there would be no joy in listening and discussing tradition" (Zohar I, 138a).[29] In fact, Targum Pseudo-Jonathan on

Genesis 2:7 says that "the Lord God created with two *yeṣers*,"[30] suggesting that God is directly responsible for the *yeṣer ha-ra'*.

Bereishit Rabbah 9:7 is even more explicit in making God responsible for the *yeṣer ha-ra'*, explaining that God's declaration that His creation is "very good" in Genesis 1:31 was in response to the operation of the *yeṣer ha-ra'*. The reason offered is that, "without the *yeṣer ha-ra'*, no man would ever build a house, or marry a woman, or beget offspring, or conduct commerce."[31]

Yoma 69b provides an especially vivid example of this reasoning, describing an incident that occurred in the days of Zechariah the prophet. After three days of fasting, God delivered up the *yeṣer* of idol worship to them, in the form of a fiery lion cub that emerged from the Holy of Holies. In imitation of Zechariah 5:8, they trapped it in a vessel with a leaden lid, and suddenly the old inclination towards idol worship was stripped from Israel. The sages were so impressed with the results that they prayed that the *yeṣer* of transgression (in this text identified with libido) be delivered to them in the same way. On the advice of Zechariah, they imprisoned this *yeṣer*, and the results were disastrous; with no urge to reproduce, the chickens even refused to lay eggs. Consequently, the sages blinded the *yeṣer* of transgression and set it free, so that it could continue to do its work, albeit with handicaps.[32]

## 12. The *Yeṣer Ha-Ra'* and the Pauline *Sarx*

More typically, the Babylonian Talmud develops the idea of the *yeṣer ha-ra'* alongside the good inclination (*yeṣer ha-ṭōb*) in a particular way: "Rabbi Nachman, bar Rabbi Chisda preached this: 'Why is, 'And the LORD God formed (וַיִּיצֶר) the man' written with two *yods*? Because the Holy One, Blessed Be He, created two inclinations. One is the good inclination (*yeṣer ṭōb*) and one is the evil inclination (*yeṣer ra'*)" (BT Berakhot 61a) (See [Cohon 1987](), pp. 247–48). But Rabbi Nachman, bar Chisda's interpretation, was opposed by other rabbis. So, in the same passage, he is attacked by another Rabbi Nachman, this one the son of Isaac: "But if that is the case, does not an animal, of whom it is not written concerning it that the LORD God 'formed' it, have an inclination? And do we not see that it causes damage and bites and kicks? Rather, as Rabbi Shimeon, the son of Pazzi taught . . . , 'Woe unto me from the One Who formed me (*Yoṣrî*) and woe unto me from my inclination (*yiṣrî*)!'"

Rabbi Nachman bar Isaac provides an understanding of the *yeṣer ha-ra'* that in some ways parallels St. Paul's teachings about the *sarx*, i.e., "the flesh". It is evident, for one thing, that he does not want to attribute the existence of this inclination to God's creative activity. Moreover, the evil inclination, according to Rabbi Nachman bar Isaac, has bestial qualities. It is not immediately subject to the rational, spiritual aspect of humanity that we generally identify with the *imago Dei*. But most importantly, Nachman bar Isaac recognizes a struggle in our will between obeying God and obeying this evil inclination. Whichever one we choose to serve, we can expect to be troubled by the one which we deny. I am immediately reminded of these words from Paul, in the fifth chapter of his Epistle to the Galatians: "16 But I say, walk by the Spirit, and do not gratify the desires of the flesh. 17 For the desires of the flesh are against the Spirit, and the desires of the Spirit are against the flesh; for these are opposed to each other, to prevent you from doing what you would". The *yeṣer ha-ra'* and the *sarx* can be viewed as analogous to one another, then, although it is probably foolhardy to equivocate them.[33]

The problem is that rabbinic Judaism does not really provide an origin story for *yeṣer ha-ra'* in the same way that Paul seems to do for his doctrine of the *sarx*. As Samuel S. Cohon observed, the early chapters of Genesis "served the Rabbis as Biblical support for their doctrine of the Yezer, but they establish no connection between the sin of Adam and the disposition to evil."[34]

### 13. Original Sin Disappeared with Apocalypticism from Judaism

As I draw this paper to a conclusion, I want to speculate just a bit as to why the story of Adam and Eve and their fall from grace virtually disappears from rabbinic discussions of the problem of evil.

I think that this is by and large the result of the systematic eradication of apocalypticism from Jewish religious life in the decades following the disastrous Bar-Kochva revolt. As we have observed, original sin is actually an apocalyptic doctrine. And so, when the rabbis began their project of reshaping Judaism around the lived praxis of obedience to the *mitzvot*, to the exclusion of apocalyptic expectation, the analogues to Paul's doctrine of original sin in a rabbinic milieu simply withered on the vine. Of course, this also proved convenient in that it deflected attention from the Christian Messiah, whom the Church had quite convincingly portrayed as the great Champion who would redeem humanity from the effects of this original sin.

In the end, Judaism became more this-worldly, focusing on the lived experience of Torah observance in the day-to-day. The Fall became simply one among many other historical events in the story of Israel. But for Christianity, with its sacramental perspective, the Fall took on a more cosmic character, setting up the entire plot of salvation history. This sweeping historical scope is ultimately a part of Christianity's apocalyptic heritage.

Apocalyptic texts function differently than carefully worded expositions of systematic theology. Much of their power is in direct correlation to their imprecision. For example, the "abomination that causes desolation" meant something specific for the author of Daniel and 1 Maccabees, but Matthew and Mark could use the same expression in their predictions of the destruction of the Temple for something similar, but completely different, and readers in the generations that have followed have continued to speculate in regards to other applications of these texts. What if Paul is doing something more like this in Romans five?

Perhaps the greatest benefits to a recovery of original sin as an apocalyptic doctrine are to be immediately achieved in settings where Christians from various traditions are actively engaging one another in conversations about what unites and what divides us.[35] It is imperative for any debate about original sin to get a good grasp of what our various proof-texts meant for their original authors and audience, thus providing a more proper historical and theological context for even beginning to talk about original sin with one another. Reading Romans five as an apocalyptic text might shake up our discussions with one another in just the right way.

**Funding:** This research received no external funding.

**Institutional Review Board Statement:** Not applicable.

**Informed Consent Statement:** Not applicable.

**Data Availability Statement:** Not applicable.

**Conflicts of Interest:** The author declares no conflict of interest.

### Notes

[1] See, for instance, (Campbell 1871, pp. 28–29), where he argues against the terminology of "original sin" as unbiblical, but still maintains that in Adam our nature sinned.

[2] Oliver Crisp is surely correct when he writes, "There is no single, agreed-upon definition of original sin in the Christian tradition—no hamartiological analogue to the famous 'definition' of the person of Christ given in the canons of the Council of Chalcedon," (Crisp 2015, p. 256). He does admit boundaries for the doctrine, however, and I find these helpful for our purposes: "All versions of the doctrine that are theologically orthodox must avoid the heresy of Pelagianism, according to which human sinfulness is a matter of imitation not imputation, and is not in principle a foregone conclusion for any particular individual. Also to be avoided is the error of semi-Pelagianism, according to which humans beings are able to exercise their free will independent of divine grace in order to co-operate with divine grace in bringing about their own salvation". Crisp's purpose in this article is to account for the classical Reformed doctrine of original sin in light of the scientific evidence for evolution. He is especially keen to emphasize that original sin must not be construed as implying original guilt, as some Reformed theology is accustomed to argue (pp. 259–60).

He confesses that part of his motivation is to find common ground with Catholics and Orthodox, who interpret original sin or ancestral sin as privative in nature, stressing the loss of original beatitude (pp. 264–65).

3    Stanley Porter identifies the interruption of a hypothetical interlocutor here, and does not believe that Paul resumes his original point until verse 19 (S. E. Porter 1991, p. 672).

4    Some commentators (see Alford 1958, p. 364) argue that this should be translated "by means of one trespass".

5    (Fitzmyer 1993, p. 421). Stanley Porter makes the case that the structure of these verses is influenced by Greek rhetoric, constituting, in fact, a diatribe (S. E. Porter 1991, p. 668). Specifically, here Paul employs an exemplum, "which in diatribe often singles out a mythical or ancient hero to exemplify particular moral qualities or attributes," (pp. 669–70). He also observes that the "short, crisp sentences" that Paul uses here "are typical of diatribe," (p. 671), as is the "parallel opposition" built up between Adam and Jesus, (pp. 672, 674). The shift to third person for the exempla of Adam and Jesus "conforms well to diatribe style," (p. 676).

6    See, for instance, (Singer 2017), where he vehemently states that "there is nothing in the Eden narrative that could be construed as support for Paul's teaching on humanity's dire condition". See also (Edersheim 1886, p. 165; Cooper 2004, p. 445).

7    Author's translation. See (Cohon 1987, pp. 255–56).

8    (Cohon 1987, p. 271). But he also writes that Judaism's "moral realism kept it from the quagmires into which Pauline Christianity fell".

9    As did Charles Spurgeon on this verse in his *Treasury of David* (Spurgeon 1988). Samuel S. Cohon dismissed this interpretation, insisting that the psalmist "does not suggest the sinfulness of the act of generation, but rather the general instability of the race of humans, who are prone to sinfulness from the very womb". For Cohon, this is corroborated in the Psalm itself when the psalmist begs for the restoration of his morally pure state (Cohon 1987, p. 226). The phrase "sinfulness of the act of generation" suggests that Cohon believed that the Christian doctrine of original sin is necessarily Augustinian.

10    See also the parallels in Exodus 20:5–6 and Deuteronomy 5:9–10, and Samuel S. Cohon's commentary on these in (Cohon 1987, p. 254), and especially p. 257: "In view of the solidarity of the Jewish people, each generation completely identifies itself with the preceding ones and assumes responsibility for their misdeeds".

11    (Dunn 1988, p. 272). But he argues that "his theme is original *death* more than original *sin*," (p. 273). See also (Cooper 2004, p. 447): "I am inclined to the simpler view, propounded almost a century ago by Israel Lévi, that during the rabbinic period, contrary to the mainstream opinion, some Jews had a notion of original sin that included the idea that the first sin was transmitted from Adam and Eve to their descendants".

12    See Notes 7 above.

13    Fitzmyer mentions several of these Second Temple period works in his commentary on Romans 5, and appeals to them as evidence that Paul was drawing upon an established, Jewish tradition, (Fitzmyer 1993, p. 413). This translation of 2 Baruch is from (Klijn 2016). Sanday and Headlam said as much nearly 100 years prior in their commentary, as well (Sanday and Headlam 1895, p. 134).

14    (Klijn 2016, p. 619). In contrast, Samuel S. Cohon believed that 2 Baruch was composed as a polemical response to "the notions of human sinfulness set forth in IV Ezra and possibly in the Epistles of Paul," (Cohon 1987, p. 233).

15    See Notes 7 above.

16    Translation from (Metzger 2016).

17    Porter observed that the "lines of escape form the problem of the evil heart" in 4 Ezra "are precisely like the rabbinical treatment of the evil *yeçer*," (F. C. Porter 1901, p. 151).

18    This passage is taken from the RSV.

19    Porter insists that "the evil heart explains Adam's sin, but is not explained by it," (F. C. Porter 1901, p. 147). But the text seems far more ambiguous to me.

20    (Metzger 2016, p. 521). See also (Cohon 1987, p. 232; Charles 1896, lxx; F. C. Porter 1901, p. 146). (But Porter identified the *cor malignum* with the *yeṣer ha-raʿ*!).

21    Translations from the Mishnah and Babylonian Talmud are by the author.

22    On this and related texts, see (Cohon 1987, p. 246).

23    See Notes 7 above.

24    Consult Fine (2023). Alan Cooper argues that "medieval Judaism developed its own versions of the doctrine of original sin, products of varying combinations of internal development and Christian influence," (Cooper 2004, p. 447). Noting the renewed stress on Augustinian interpretations of the Fall that emerged during the Reformation and the Counter-Reformation, Cooper observes that "a dour view of human nature, rooted in original sin . . . seems to have spread throughout Northern Europe in the sixteenth and seventeenth centuries, and cannot have failed to make an impression on Jewish thinkers," (p. 450).

25    Cohon (1987, p. 220). He gives special attention to the Zoroastrian influence on First Enoch on p. 230 and on Wisdom 2:24 on p. 231. (He actually argues that this passage in Wisdom owes its inspiration to elements that appear in First Enoch).

26    Author's translation. See also (Cranfield 1975, p. 288).

27  See (Cohon 1987, pp. 229–30), where he holds these verses up as a corrective counter-balance to Ecclesiasticus 25:24.

28  See (Schechter 1909, p. 243). He finds more biblical background in Genesis 6:5 and 8:21, Deuteronomy 31:21, and Psalm 103:14, p. 242. He also warns against a precise definition of *yeṣer ha-ra'*: "the term is so obscure and so variously used as almost to defy any real definition," (p. 242).

29  See Notes 7 above.

30  See Notes 7 above.

31  See Notes 7 above.

32  On this and other examples, see (Cohon 1987, p. 248). But it seems to me that he forces the interpretation of many of the texts to which he appeals, ignoring a joyous expectation in most of them that humanity will one day be delivered from *yeṣer ha-ra'* in a conclusive way. Instead, Cohon concludes that *yeṣer ha-ra'* is "in reality neutral," and that "man can use his passional nature for good as well as for evil". See also (Edersheim 1886, p. 167), and Sanday and Headlam's response to him (Sanday and Headlam 1895, p. 137).

33  This possibility was already explored, and rejected, in (F. C. Porter 1901, pp. 93–107). But he made, it seems to me, the mistake of treating the *yeṣer ha-ra'* as though it was a well-defined concept in rabbinic thought, used systematically and with consistency. My contention here is that Berakhot 61a is using *yeṣer ha-ra'* differently than many of the other texts assembled. Porter was forced to acknowledge that this talmudic text is a bit of an outlier, because unlike other rabbinic texts, the *yeṣer ha-ra'* here seems to be alien to the original purpose of the Creator (F. C. Porter 1901, p. 117). Porter also errs in forcing a Platonic dualism on Paul. *Contra* Porter, I do not believe that Paul uses *sarx* as an essentially corporeal term in opposition to *pneuma*. Unfortunately, Porter construes *sarx* as equivalent to our word "body" (F. C. Porter 1901, p. 134).

34  (Cohon 1987, p. 225). But see Rashi on Genesis 2:25: "Adam had no *yeṣer ha-ra'* until he ate from the tree and the *yeṣer ha-ra'* entered into him and he knew the difference between good and evil", author's translation. See also Cohon's treatment of this (Cohon 1987, p. 247).

35  The original version of this paper was presented in just such a setting, at the 2022 *Ad Fontes* conference sponsored by the Eighth Day Institute and Gerber Institute of Catholic Studies in Wichita, Kansas.

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
