# Peer review of "Yeṣer ha-Raʻ and Original Sin"

_religions, doi:10.3390/rel14060733_

Round 1

Reviewer 1 Report

This is a short article, and while it is well-written and demonstrates a good knowledge of the primary literature, it does not appear to know the secondary literature well, or the scholarship on original sin. Indeed, the author frames the article as a rejection of his rabbi friend's opinion, one popularly shared, that original sin has no Jewish precedents; but scholars know this well. See, eg, Samuel Cohon, "Original Sin," Hebrew Union College Annual Vol. 21 (1948), pp. 275-33, the opening sentence of which states that Original Sin is as Jewish as it is Christian.

There are other references which could be made - more recent scholarship has located Original Sin re-emerging in the Lurianic Kabbalah, along with several other apparently Christian motifs (Shaul Magid, From Metaphysics to Midrash: Myth, History, and the Interpretation of Scripture in Lurianic Kabbala (Bloomington: Indiana University Press, 2008) esp. pp. 34–74. Alan Cooper, ‘A Medieval Jewish Version of Original Sin: Ephraim of Luntshits on Leviticus 12’, HTR 97.4 (2004), pp. 445–59).

The question then is, why does Original Sin tend to appear along with a set of certain other (often messianic-apocalyptic) motifs? To rewrite the article in this direction, or another, is possible, depending on the willingness of the author, and this could make it a worthwhile contribution. But as it stands it is not ready for publication in an academic journal. Of course, if the author's intention is to "help Christians to be better partners in dialogue with their Jewish friends and neighbors," then such a journal is not the right place for this article.

Author Response

Thank you so much for your thoughtful guidance.

The article is significantly longer.

Added “Cohon, Samuel S. 1987. Essays in Jewish Theology. Cincinnati: Hebrew Union College Press.” to list of references. Included his thoughts on original sin, Psalm 51:7, Second Temple period literature, and yeá¹£er ha-raÊ»  in the article. 

Added “Sanday, W. and Headlam, A.C. 1895. A Critical and Exegetical Commentary on the Epistle to the Romans. New York: Charles Scribner’s Sons, The International Critical Commentary.” to list of references. Included thoughts on Romans 5 in the article.

Added “Edersheim, Alfred. 1886. The Life and Times of Jesus the Messiah, New American ed. New York: E. R. Herrick & Company, vol. 1.” to list of references.

Added “Porter, Frank Chamberlin. 1901. The Yeçer Hara: A Study in the Jewish Doctrine of Sin. In Biblical and Semitic Studies. New York: Charles Scribner’s Sons, Yale Bicentennial Publications, pp. 93-156.” to list of references.

Added “Charles, R. H. 1896. The Apocalypse of Baruch Translated from the Syriac. London: Adam and Charles Black.” to list of references.

Added “Porter, Stanley E. 1991. The Argument of Romans 5: Can a Rhetorical Question Make a Difference? JBL 10/4, pp. 655-77.” to list of references. Included thoughts on Romans 5 as diatribe.

Added “Cranfield, C. E. B. 1975. A Critical and Exegetical Commentary on the Epistle to the Romans. Edinburgh: T. & T. Clark, vol. 1, The International Critical Commentary.” to list of references. Included thoughts on Romans 5 in the article.

Added “Dunn, James D. G. 1988. Romans 1-8. Grand Rapids: Zondervan, Word Biblical Commentary 38A.” to list of references. Included thoughts on Romans 5 in the article.

Added “Crisp, Oliver D. 2015. On Original Sin. International Journal of Systematic Theology 17/3, pp. 252-66.” to list of references.

Added “Fine, Lawrence. Tikkun in Lurianic Kabbalah. Available online: https://www.myjewishlearning.com/article/tikkun-in-lurianic-kabbalah/ (accessed on 20 May 2023).” to list of references.

Added “Cooper, Alan. 2004. A Medieval Jewish Version of Original Sin: Ephraim of Luntshits on Leviticus 12. HTR 97/4, pp. 445-59.” to list of references.

Included much more primary source material.

Added these three paragraphs to better explain my intent with all of this:

But Paul does not owe only his doctrine of the fall to this apocalyptic framework. His teachings on the destiny of humanity redeemed from this fall are also apocalyptic. In Romans 5:17, Paul writes, “If, because of one man’s trespass, death reigned through that one man, much more will those who receive the abundance of grace and the free gift of righteousness reign in life through the one man Jesus Christ.” 
This echoes an apocalyptic expectation expressed in Daniel 7:22 and 27 that in the eschaton God’s holy ones will receive an everlasting kingdom.1 It is also found in the Psalms of Solomon 3:12: “Those who fear the Lord will rise up into life eternal and their life will be in the light of the Lord and it will never again be eclipsed.”2
So, both Paul’s doctrine of original sin and the eschatological reign of the holy ones who have been redeemed from it are part of a larger corpus of Jewish apocalyptic literature. What is unique to Paul is his proposal that Jesus provides the redemption necessitated by the Fall. Consequently, I am convinced that we must approach Romans five as primarily an apocalyptic text. Most commentaries on Romans five do the opposite, treating it as a piece of systematic theology that is drawing upon a few ideas that it shares in common with a handful of apocalyptic texts.
----
1See Dunn 1988, 282.
2Author’s translation. See also Cranfield 1975, 288.

Reviewer 2 Report

The author made his point showing that the idea of the original sin does appear in ancient Jewish sources. I suggest that the author mentions Breshit Rabbah 17:8, where Jewish women are remided that they have enherited the sin of Eve. 

As for the reasons for its opfuscation, I cannot agree with the author. Although the idea of the original sin is present in early Jewish sources, it has always remained a marginal one, because it does not serve the objectives of Judaism. Since the essence of Judaism is an "existentialist" one (to live the moment, to celebrate through the commandments the everyday), the original sin is for it a "historical event.". Christianity, on the other hand, has metaphysical objectives. This explains the theological importance of the original sin in Christianity and its apocalyptic significance.  

Author Response

Thank you for your helpful comments!

The article is significantly longer.

Added “Cohon, Samuel S. 1987. Essays in Jewish Theology. Cincinnati: Hebrew Union College Press.” to list of references. Included his thoughts on original sin, Psalm 51:7, Second Temple period literature, and yeá¹£er ha-raÊ»  in the article. 

Added “Sanday, W. and Headlam, A.C. 1895. A Critical and Exegetical Commentary on the Epistle to the Romans. New York: Charles Scribner’s Sons, The International Critical Commentary.” to list of references. Included thoughts on Romans 5 in the article.

Added “Edersheim, Alfred. 1886. The Life and Times of Jesus the Messiah, New American ed. New York: E. R. Herrick & Company, vol. 1.” to list of references.

Added “Porter, Frank Chamberlin. 1901. The Yeçer Hara: A Study in the Jewish Doctrine of Sin. In Biblical and Semitic Studies. New York: Charles Scribner’s Sons, Yale Bicentennial Publications, pp. 93-156.” to list of references.

Added “Charles, R. H. 1896. The Apocalypse of Baruch Translated from the Syriac. London: Adam and Charles Black.” to list of references.

Added “Porter, Stanley E. 1991. The Argument of Romans 5: Can a Rhetorical Question Make a Difference? JBL 10/4, pp. 655-77.” to list of references. Included thoughts on Romans 5 as diatribe.

Added “Cranfield, C. E. B. 1975. A Critical and Exegetical Commentary on the Epistle to the Romans. Edinburgh: T. & T. Clark, vol. 1, The International Critical Commentary.” to list of references. Included thoughts on Romans 5 in the article.

Added “Dunn, James D. G. 1988. Romans 1-8. Grand Rapids: Zondervan, Word Biblical Commentary 38A.” to list of references. Included thoughts on Romans 5 in the article.

Added “Crisp, Oliver D. 2015. On Original Sin. International Journal of Systematic Theology 17/3, pp. 252-66.” to list of references.

Added “Fine, Lawrence. Tikkun in Lurianic Kabbalah. Available online: https://www.myjewishlearning.com/article/tikkun-in-lurianic-kabbalah/ (accessed on 20 May 2023).” to list of references.

Added “Cooper, Alan. 2004. A Medieval Jewish Version of Original Sin: Ephraim of Luntshits on Leviticus 12. HTR 97/4, pp. 445-59.” to list of references.

Included much more primary source material, including part of Bereshit Rabbah 17:8.

Added these three paragraphs to better explain my intent with all of this:

But Paul does not owe only his doctrine of the fall to this apocalyptic framework. His teachings on the destiny of humanity redeemed from this fall are also apocalyptic. In Romans 5:17, Paul writes, “If, because of one man’s trespass, death reigned through that one man, much more will those who receive the abundance of grace and the free gift of righteousness reign in life through the one man Jesus Christ.” 
This echoes an apocalyptic expectation expressed in Daniel 7:22 and 27 that in the eschaton God’s holy ones will receive an everlasting kingdom.1 It is also found in the Psalms of Solomon 3:12: “Those who fear the Lord will rise up into life eternal and their life will be in the light of the Lord and it will never again be eclipsed.”2
So, both Paul’s doctrine of original sin and the eschatological reign of the holy ones who have been redeemed from it are part of a larger corpus of Jewish apocalyptic literature. What is unique to Paul is his proposal that Jesus provides the redemption necessitated by the Fall. Consequently, I am convinced that we must approach Romans five as primarily an apocalyptic text. Most commentaries on Romans five do the opposite, treating it as a piece of systematic theology that is drawing upon a few ideas that it shares in common with a handful of apocalyptic texts.
----
1See Dunn 1988, 282.
2Author’s translation. See also Cranfield 1975, 288.

Also added this paragraph, towards the end:

In the end, Judaism became more this-worldly, focusing on the lived experience of Torah observance in the day-to-day. The Fall became simply one among many other historical events in the story of Israel. But for Christianity, with its sacramental perspective, the Fall took on a more cosmic character, setting up the entire plot of salvation history. This sweeping historical scope is ultimately a part of Christianity’s apocalyptic heritage.

Round 2

Reviewer 1 Report

It's much better now. Thank you for your willingness to incorporate additional material.